# Graphene Quantum Dots as Intracellular Imaging-Based Temperature Sensors

**DOI:** 10.3390/ma14030616

**Published:** 2021-01-29

**Authors:** Bong Han Lee, Ryan Lee McKinney, Md. Tanvir Hasan, Anton V. Naumov

**Affiliations:** 1Department of Physics and Astronomy, Texas Christian University, Fort Worth, TX 76129, USA; bong.lee@tcu.edu (B.H.L.); ryan.l.mckinney@tcu.edu (R.L.M.); tanvir.hasan@tcu.edu (M.T.H.); 2Biosystems and Biomaterials Division, National Institute of Standards and Technology, Gaithersburg, MD 20899, USA

**Keywords:** graphene quantum dots, nanothermometry, fluorescence, in vitro, temperature sensing

## Abstract

Non-invasive temperature sensing is necessary to analyze biological processes occurring in the human body, including cellular enzyme activity, protein expression, and ion regulation. To probe temperature-sensitive processes at the nanoscale, novel luminescence nanothermometers are developed based on graphene quantum dots (GQDs) synthesized via top-down (RGQDs) and bottom-up (N-GQDs) approaches from reduced graphene oxide and glucosamine precursors, respectively. Because of their small 3–6 nm size, non-invasive optical sensitivity to temperature change, and high biocompatibility, GQDs enable biologically safe sub-cellular resolution sensing. Both GQD types exhibit temperature-sensitive yet photostable fluorescence in the visible and near-infrared for RGQDs, utilized as a sensing mechanism in this work. Distinctive linear and reversible fluorescence quenching by up to 19.3% is observed for the visible and near-infrared GQD emission in aqueous suspension from 25 °C to 49 °C. A more pronounced trend is observed with GQD nanothermometers internalized into the cytoplasm of HeLa cells as they are tested in vitro from 25 °C to 45 °C with over 40% quenching response. Our findings suggest that the temperature-dependent fluorescence quenching of bottom-up and top-down-synthesized GQDs studied in this work can serve as non-invasive reversible/photostable deterministic mechanisms for temperature sensing in microscopic sub-cellular biological environments.

## 1. Introduction

As the cell research advances towards closely studying intracellular mechanisms, so does the need for monitoring the dynamics of cells in real time. Biological cell processes are governed by chemical reactions, which are fundamentally affected by temperature [1,2]. These reactions within the cell can occur in different organelles, causing varying temperature gradients to be present at one time due to processes ranging from protein expression to expelled heat from tumors [3,4]. Thus, temperature distribution within a living cell can give information of the thermodynamic functions and intracellular expressions [5], while its knowledge can prove to be essential for multiple in vitro studies.

Nanothermometry aims to identify the temperature of the specimen with sub-micrometric spatial resolution [6]. This technique of temperature assessment is used in varying systems such as photonic devices, micro-electronics, and biological cells [6,7]. In the latter, general functions of organelles such as hydrolysis of glucose for adenosine triphosphate (ATP) generation [8] can vary as their processes and dynamics rely heavily upon temperature. On the other hand, cancer tumors can express heat production through increased metabolic rates [9], glucose and pyruvate combustion releases of free energy through cellular respiration [10], and intracellular ion exchange [11]. Therefore, to study these processes on the microscale, system-specific and stable nanothermometers are required. Although several techniques have been developed involving nanomaterials, the needs of biological applications still require a highly deterministic non-invasive biocompatible approach allowing for intra- and extracellular temperature measurements. Nanothermometers can come in the forms of organic dyes, quantum dots, inorganic nanocomposites, and a variety of nanomaterials-based constructs [12,13]. In these, thermometry mechanisms rely on a change of detectable properties of the nanomaterial with temperature. A widely utilized luminescence nanothermometry [14,15,16] relies on temperature-dependent, but otherwise stable optical properties of the nanomaterial. This method encompasses intensity, band shape, spectra, polarization, bandwidth, and lifetime nanothermometry approaches relying on the variation of corresponding properties [6]. In the most common, intensity luminescence nanothermometry (ILN), the fluorescence intensity change is observed to indicate the temperature of the local system. Luminescence nanothermometry has been used so far in integrated photonic devices and micro/nanoelectronics [17,18,19]. Furthermore, it has been extensively utilized in medicine with the spatial resolution allowing us to detect the location of the sensor for highly localized temperature readings [20,21]. 

A drawback to using nanomaterials intracellularly is their accumulation-derived toxicity [22] and immunogenicity [23]. While they can be non-toxic in small doses, many nanoparticles do not easily excrete from the body, eventually accumulating to toxic levels [24]. Chemical modifications of nanoparticle surface allow for more biocompatible as well as less immunogenic solutions [25]. Quantum dots (QDs), for instance, one of the most commonly utilized classes of nanoparticles, have a diversity of chemical and optical properties due to the possibility of altering their surface chemistry and size [26]. In previous studies, CdTe/ZnS QDs [15], CdSe/ZnS QDs [27], and CdTe QDs [28] were used as ILNs detecting minor temperature changes. These structures positioned also for bioapplications are biocompatible generally at low concentrations. In separate studies, CdTe/ZnS QDs are found to have a cell survival ˃80% from 1 to 7 μM [29], while CdSe/ZnS QD show the same cell survival from 1.2 to 3.7 μg/mL [30]. However, at concentrations above 3.7 μg/mL those cells’ viability drops below 20%, and CdTe QDs have a cell survival rate of ~30% at concentrations of only 300 nM [31]. These high toxicity profiles call for the development of more biocompatible alternatives. PEG-elation of these inorganic GQDs adds an extra complicating synthetic step and still retains a danger of accumulation-derived toxicity especially with coating degradation.

Carbon nanomaterials may present yet another perspective class of nanothermometers due to the environmental sensitivity of their intrinsic optoelectronic properties as well as high structural tunability via a variety of synthetic approaches. One of the basic carbon nanomaterials, graphene is applied in many scenarios due to its high thermal and chemical stability as well as electrical conductivity [32]. Its zero band gap electronic structure, however, does not allow for its application in the ILN [33]. Graphene quantum-dots (GQDs), being few nanometers in size and often an oxidized derivatives of graphene do not possess those disadvantages. They are substantially more suited for biosensing [17], bioimaging [34,35], and drug delivery applications due to their biocompatibility, water solubility, small size, and quantum confinement-defined intrinsic fluorescence [29]. Unlike multiple organic dyes used for ILN [6], GQDs can be rendered to exhibit high yield but photostable emission, are simple in synthesis, biocompatible [33], and biodegradable [36]. Several types of GQDs synthesized via different routes have been successfully utilized for ILN [35,37] applications utilizing their temperature-induced fluorescence quenching. However, the biocompatibility of these platforms at higher doses has not been explored as well as their NIR quantum yields, while their fluorescence in the visible is not ideal for the majority of in vivo applications often hampered by tissue scattering and absorption.

To improve on these advances and develop nanothermometers with the most desired properties for biotechnology, we produce two different types of highly biocompatible GQD ILNs: Nitrogen-doped graphene quantum-dots (N-GQDs) prepared via bottom-up synthesis from glucosamine precursor and reduced graphene oxide-derived quantum dots (RGQDs) synthesized top-down from reduced graphene oxide (rGO). N-GQDs exhibit high (over 60%) yield fluorescence in the visible, while RGQDs fluoresce both in the visible and near-infrared (NIR). Due to high near-infrared light tissue penetration depth, the NIR emission of these novel nanoplatforms can be utilized for ex vivo and select in vivo fluorescence applications [38]. Ideally, such emission is desired to be within the first (750–900 nm) and second (1000–1700 nm) biological windows for reduction of scattering and absorbance of light by biological tissue, allowing imaging with over 2 cm of tissue penetration depth [39]. So far, there has been no highly fluorescent NIR emitting GQDs near the second biological window to be used as nanothermometers [35,40]. In order to effectively monitor intracellular processes, it is required that the system must be as undisturbed as possible through the application of the nanothermometer. Based on their size and over 1 mg/mL biocompatibility as well as interdependence of their optical, electronic, and thermal properties arising from the graphene core, we expect N-GQDs and RGQDs to fully satisfy requirements for effective nanothermometry platforms, as their performance is evaluated in vitro.

## 2. Materials and Methods

### 2.1. Sample Synthesis 

GQDs were synthesized using both the bottom up and the top down approaches. N-GQDs were synthesized bottom-up using microwave-assisted hydrothermal treatment following a previously developed protocol [33]. In brief, an aqueous solution of glucosamine-HCl (346299, Sigma Aldrich, St. Louis, MO, USA) was processed in a microwave (HB-P90D23AP-ST, Hamilton Beach, Southern Pines, NC, USA) for 60 min at 270 W. During the treatment, glucosamine-HCl was polymerized forming nanosized quasi-spherical graphitic N-GQDs. The sample was then purified from unreacted precursor using 1 kDa bag dialysis (Spectrum Chemical Mfg. Corp., New Brunswick, NJ, USA) for 2 h against DI water that was replaced every 30 min. RGQDs were synthesized using top-down UV-assisted chemical oxidation/exfoliation of rGO using NaClO. rGO (HC-380190900000, Graphene Supermarket, Ronkonkoma, NY, USA) was first suspended in water at a concentration of 0.20 mg/mL and ultrasonically treated (Q55-100, QSonica, Newtown, CT, USA) for 60 min at 22 W to dissipate large aggregates. NaClO, 5% w/v, (ACSLC246301, LabChem, Zelienople, PA, USA) was added to the 20 mL aqueous suspension of rGO and was irradiated with a 302 nm benchtop UV transilluminator (LMS-20, Daigger Scientific, Hamilton, NJ, USA) for 120 min at 8 W to enhance the reaction rate. The product was further purified from oxidative debris using 1 kDa dialysis bag for 24 h against water replaced every 30 min in the first 3 h. Then it was filtered using a 0.22 μm hydrophilic syringe filter (SimPure, Bellevue, WA, USA) to remove unreacted rGOs and concomitantly sterilize the sample.

### 2.2. Insulated Thermal Microscopy Device (ITMD)

To heat the cells for ILN testing, an insulated thermal microscopy device was engineered in house using commercially purchased Peltier thermoelectric modules (TE Technology, Inc., Traverse City, MI, USA). The modules were fastened to a customized copper plate in acrylic housing, with a K-type thermocouple. Cover slips seeded with cells were placed onto the plates of the device allowing cell sample heating/cooling (Appendix A). A tunable DC power supply was used to apply potential to the thermoelectrical modules to control heating of the system. For bulk measurements, a similar thermoelectric device was constructed allowing the variation of temperature of the cuvette inside the spectrophotometer (Cary 60 UV-Vis, Agilent, Santa Clara, CA, USA) and spectrofluorometer (Nanolog, HORIBA Scientific, Edison, NJ, USA).

### 2.3. Optical Characterization

The absorbance of the samples was measured using the Cary 60 UV-Vis spectrophotometer in the range of 200–1000 nm. The fluorescence was monitored using the spectrofluorometer with visible CCD deep cooled camera (Syncerity-1024x256-OE, HORIBA Scientific, Edison, NJ, USA) and NIR InGaAs arrays LN_2_ cooled camera (Symphony II, HORIBA Scientific, Edison, NJ, USA) for NIR detection. N-GQDs/RGQDs were excited with a 450 W broadband xenon lamp of Horiba Nanolog (HORIBA Scientific, Edison, NJ, USA) at 400 nm excitation wavelength as their emission was measured in the visible range of 420 nm to 762 nm. NIR fluorescence was measured with 475 nm lamp excitation and emission collected in the range of 850 nm to 1500 nm. Fluorescence and absorbance were measured as the temperature of the samples was varied in situ from 27 °C to 49 °C and back in 2 °C temperature increments.

### 2.4. Cell Work

HeLa cells (Human cervical carcinoma) (ATCC, Manassas, VA, USA) were utilized for in vitro experiments. These cells were cultured in an incubator (Midi CO_2_, Thermo Scientific, Waltham, MA, USA) at 37 °C with 5% CO_2_ in DMEM (D6046, Sigma-Aldrich, St. Louis, MO, USA) supplemented with 10% FBS (16140-063, Gibco, Dublin, Ireland), L-Glutamine (G7513, Sigma-Aldrich, St. Louis, MO, USA), Non-essential Amino Acid solution (M7145, Sigma-Aldrich, St. Louis, MO, USA), and 1% Penicillin/Streptomycin (P4333, Sigma-Aldrich, St. Louis, MO, USA).

To assess the nanothermometry capabilities at the microscale, 10,000 cells have been seeded into each well of the 6-well plate with coverslips at the bottom. After 24 h enabling cell attachment, aqueous suspensions of N-GQDs and RGQDs were added to each well to render a final GQD imaging concentration of 1 mg/mL and incubated for another 24 h. Cell medium was removed from each well and 1X phosphate buffered saline (PBS, Bioland Scientific LLC, Baltimore, MD, USA) solution was added and removed to wash off the remaining medium. 4% Formaldehyde (28908, Thermo Scientific, Waltham, MA, USA) diluted in 1X PBS was added to each well and left for 30 min at 4 °C for fixation. The 4% formaldehyde solution was further removed from each well, and 1X PBS was added to preserve the cells on the slides. A drop of 1X Fluoromount-G^TM^ mounting medium (00-4958-02, Invitrogen, Carlsbad, CA, USA) was added to each slide and the cover slip with cells well-fixed onto it, after which slides/cover slips were sealed with nail polish for imaging.

### 2.5. Microscopy

A semi-motorized inverted microscope (IX73P2F, Olympus, Center Valley, PA, USA) with IR-corrected UPLANAPO 60x/0.90na objective (1-UB831, Olympus, Center Valley, PA, USA) and visible/NIR spectrally-resolved imaging capability was utilized for microscopy imaging. The visible part of the setup was coupled to DSU disk-spinning unit for confocal imaging (Olympus, Center Valley, PA, USA) and further to a CMOS (Prime 95B, Photometrics, Tucson, AZ, USA) camera. NIR pathway was coupled to an InGaAs FPA (ZephIR^TM^ 1.7, Photon etc., Montreal, Quebec, Canada) camera via Photon etc. Hyperspectral imager allowing spectrally-resolved image collection in the NIR. Visible imaging utilized 460 ± 20 nm filtered lamp excitation and 535 ± 20 nm filtered emission collection, while NIR emission was excited with 808 nm laser with hyperspectral emission collection in the range of 900–1600 nm.

### 2.6. Image Analysis

ImageJ software (1.53a, National Institutes of Health, Bethesda, MD, USA) was used to assess the intracellular fluorescence level. To quantify the fluorescence, the corrected total cell fluorescence (CTCF) was calculated. The CTCF was found by first outlining the cell of interest and their respective background. These measurements then generated the selected area and integrated density for the cell, and mean intensity for the background. The product between the mean fluorescence background and the outlined cell area subtracted from the integrated density yielded the CTCF that was considered as a quantitative measure of GQD fluorescence within the cells.

## 3. Results and Discussion

Two types of graphene quantum dots, N-GQDs and RGQDs, are synthesized via the bottom up and top down approach, respectively, are utilized in this work. N-GQDs are synthesized from an aqueous solution of glucosamine-HCl undergoing one-step hydrothermal reaction using a commercially available microwave [33]. For the production of RGQDs via the top down approach, an aqueous solution of micrometer-sized rGO flakes is treated with NaOCl, and exposed to UV light to accelerate the reaction rate allowing for oxidation and scission of rGO [41]. Transmission electron microscopy (TEM) performed to characterize these nanomaterials in this (Appendix A) and previous works indicates the average size of N-GQDs and RGQDs of 5.5 nm and 3.5 nm, respectively, and confirms their graphitic lattice structure and spacing [33,41,42]. The structures of N-GQDs and RGQDs are also confirmed via Raman spectroscopy with observable D and G bands (Appendix A), representing the sp^2^ hybridized carbon and disordered structure, respectively [33,41]. Both RGQDs and N-GQDs are well-characterized and possess distinct optical signatures in absorption (Appendix A) and fluorescence (Appendix A) that can be utilized effectively for intensity luminescence nanothermometry. N-GQD absorption spectra possess a characteristic feature at 250 nm attributed to the $\pi -\pi^{*}$ transition at the C=C bond, and a slightly broader peak at 306 nm peak attributed to the $n-\pi^{*}$ transitions at C=O and C=N bonds, confirmed by the Fourier transform infrared (FTIR) spectra [33]. RGQDs, on the other hand, only exhibit the blue-shifted 230 nm $\pi -\pi^{*}$ transition feature at the C=C bonds. As the suspension temperature is increased inside the absorption spectrometer in 2 °C increments, both RGQDs and N-GQDs exhibit similar changes in their absorbance spectra.

The aforementioned absorbance features of N-GQDs and RGQDs show no temperature dependence in the ultraviolet and visible regions (Figure 1a,c) suggesting no apparent changes to their major chemical structure. However, for both N-GQDs and RGQDs NIR spectra show the appearance of a minor temperature-dependent feature (insets for Figure 1a,c). Those peaks having linear intensity dependence on temperature (Figure 1b,d) show complete reversibility (Appendix A) and may arise from thermally-activated defect states suggested in our previous works in regards to N-GQD NIR fluorescence [33].

Both N-GQDs and RGQDs exhibit broad excitation-dependent visible emission attributed to the size-dependent quantum confinement effects [33,36]. Additionally, N-GQDs possess a weak NIR fluorescence originating from their surface defect states [33,36], while RGQDs exhibit substantial NIR fluorescence at ~950 nm with up to 8% quantum yields obtained from an absolute quantum yield measurement in an aqueous solution using an integrating sphere [41].

In order to explore the potential of both GQD types as ILNs, their fluorescence in aqueous suspension is continuously measured while the temperature of the suspension is varied from 25 to 49 °C (Figure 2) inside the fluorescence spectrometer. It appears that the visible features of both N-GQDs and RGQDs experience monotonic decrease with temperature down to 19.3 and 16.8%, respectively, at 49 °C. Peak intensities quench linearly with temperature (Figure 2b,d), thus providing a convenient scale for nanothermometry measurements. Additionally, the effect is reversible (Appendix A) and stable (Appendix A) as the fluorescence intensity returns back to its initial values, while the samples are cooled back to 25 °C over several cycles. For visible RGQDs emission, there is some minor hysteresis present creating deviations from the linear behavior upon cooling (Appendix A) that in conjunction with slight deviations in heating trends for RGQD visible emission (Figure 2d) potentially suggests the variation in non-radiative quenching processes originating from populating/clearing surface traps [43]. However, when considered on a full intensity scale, the heating variations do not render substantial emission changes leading to consider those as minor effects. Both linearity and reversibility make N-GQDs and RGQDs perspective candidates for ILN: Their visible emission intensities can serve as a temperature scale for consecutive measurements. The quenching effect magnitude over the 25–49 °C range is similar to that of the most effective carbon dot nanothermometers [37], however, the high visible quantum yields of up to 62% together with a well-calibrated biological temperature range present the advantage of N-GQDs and RGQDs as biological nanothermometers for in vitro and ex vivo applications. However, the advantage of the RGQDs as biological imagers is in their excitation-independent 950 nm NIR fluorescence that can be well-detected from within the animal models. Their NIR emission also exhibits quenching with temperature by up to 6.74% at 47 °C, showing high linearity in the biological 25–47 °C range with R^2^ of 0.995. This quenching also appears reversible (Appendix A) and stable (Appendix A), where during temperature decrease the linearity of RGQD emission intensity variation is still preserved in the NIR fluorescence over several cycles. This result suggests that both visible and NIR nanothermometers have a potential for a wider range of bioapplications, including some in vivo temperature detection scenarios.

Recently, Kalytchuk et al. suggested the mechanism behind this fluorescence intensity decrease due to a higher chance of obtaining non-radiative relaxation pathways from thermal effects [17,44]. Yang et al. have also denoted that hydrogen bonds and oxygen-containing surface functional groups may play a role in the decrease of the fluorescence intensity with temperature enabling temperature-dependent excitation energy traps [33,37]. This may explain the higher temperature sensitivity of the N-GQDs that are found in our works to have a high degree of functionalization [33]. Reversible temperature sensitivity been already observed with other carbon quantum dot based nanothermometers [17,35,37,44]. However, N-GQDs and RGQDs show a new variety of advantages including high biocompatibility evaluated previously via the thiazolyl blue tetrazolium bromide (MTT) and Luminescence assays, high visible quantum yields (N-GQDs) [33], simple and scalable synthesis, and potential for NIR temperature sensing (RGQDs) more applicable for in vivo studies [36,41]. 

In order to assess the nanothermometry potential of both platforms for biological applications, we explore their temperature sensitivity in vitro in HeLa cells, which have been a subject of investigation due to the need of finding ways complement treatments and diagnosis [16]. Both N-GQDs and RGQDs are known to experience successful internalization into HeLa cells due to their small size and hydrophobic carbon segments, and are biocompatible up to 1 mg/mL while still maintaining around 80% of cell viability [36,41]. Thus, both GQD types are introduced to cells at 1 mg/mL concentrations, and their temperature response is measured at a 12 h timepoint showing maximum internalization in our previous studies [36,41]. For the in vitro nanothermometry application, we test GQD response in the temperatures of physiological range (25 °C to 45 °C) [17,45]. It is achieved by heating the cover slip with HeLa cells via thermoelectric modules coupled to a copper plate in an ITMD (Appendix A) enabling fluorescence microscopy through an optical window opening and real-time temperature monitoring via a thermocouple at the cover slip set near the opening. This novel design ensured fine control over the temperature of the cell environment through providing accurate temperature readings. Similar to the trends in the spectroscopic measurements, the fluorescence intensity of N-GQDs as well as RGQDs decreases with temperature as observed in cell images (Figure 3). The fluorescence of N-GQDs in the visible and RGQDs in the visible and NIR internalized in HeLa cells is measured validating their successful internalization (Figure 4). This technique requires thorough quantification of intracellular fluorescence intensity and calibration with respect to the initial control state. It is, however, not prone to photobleaching discrepancies limiting the use of conventional fluorophores, as these GQDs are known to exhibit high fluorescence stability under irradiation [36]. Concentration-independent lifetime measurements can circumvent the need for calibration, however, they are not considered for these in vitro experiments, as carbon based nanomaterials’ lifetimes are known to be strongly affected by a number of other environmental factors including the variations in intracellular pH [46].

In order to quantify this trend, the average fluorescence intensity per unit cell area collected for over 200 cells is calculated and compared at different temperatures for both GQD types in the visible and for RGQDs in NIR. This statistical data (Figure 4) indicates even more drastic fluorescence variations than those observed spectroscopically in the visible for N-GQDs and RGQDs, with N-GQDs showing a greater response. The NIR RGQDs also exhibit the quenching expected from the prior spectral analysis and enabling temperature sensing. 

As successful fluorescence intensity-based temperature monitoring is performed, shown in this work in the visible and NIR, the potential for monitoring real-time cellular processes in terms of their biochemical reactions will be of interest [35,37,45,47], as well as temperature detection in animal models. Due to nanometer-scale dimensions, the GQD-based sensors investigated in this work can sample a variety of subcellular processes with high resolution while remaining biocompatible. Furthermore, nanothermometry can complement other modes of therapy such as hyperthermia, photothermal therapy, and even drug release [17,37,45], where N-GQDs and RGQDs can serve as promising multimodal candidates for drug delivery, imaging, and temperature sensing. 

## 4. Conclusions

In this work, we demonstrate the use of highly biocompatible novel nitrogen-doped (N-GQDs) and reduced graphene-derived (RGQDs) graphene quantum dots as promising intracellular imaging-based temperature sensors. N-GQDs (average size ~5.5 nm) and RGQDs (average size ~3.5 nm) synthesized via scalable bottom-up and top-down approaches from glucosamine and reduced graphene oxide precursors, respectively, both exhibit temperature-dependent optical properties. Although major absorption peaks of these GQDs are independent of temperature, temperature increase facilitates the appearance of a potentially defect-related state manifesting in a minor NIR absorption feature, increasing linearly with temperature. Both GQDs emit high-yield fluorescence in the visible, while RGQDs also possess emission in the near-infrared at 950 nm more optimal for in vivo studies. Their fluorescence appears to depend linearly on temperature in the biologically relevant range serving of 25 °C to 49 °C, serving as a potential mechanism for luminescence intensity nanothermometry. These capabilities are tested in vitro as N-GQDs and RGQDs are internalized into HeLa cells and show progressive decrease in fluorescence intensity as the cell temperature is raised from 25 °C to 45 °C in a pre-designed temperature-controlled fluorescence microscopy setup. This trend of intracellular fluorescence quenching is qualitatively and quantitatively assessed over the array of cells leading down to over 40% intensity decrease at 45 °C. The longer wavelength fluorescence emission of N-GQDs and RGQDs allows for low autofluorescence in vitro imaging, while RGQD NIR emission in the ~950 nm between the first and second biological windows allows for lower tissues scattering and improved penetration depth. Thus, the intrinsic linear dependence of N-GQD and RGQD fluorescence intensity on temperature allows us to probe cellular environments for a variety of thermodynamic processes in a non-invasive manner and sets the stage for in vivo nanothermometry.

## Figures and Tables

**Figure 1 materials-14-00616-f001:**
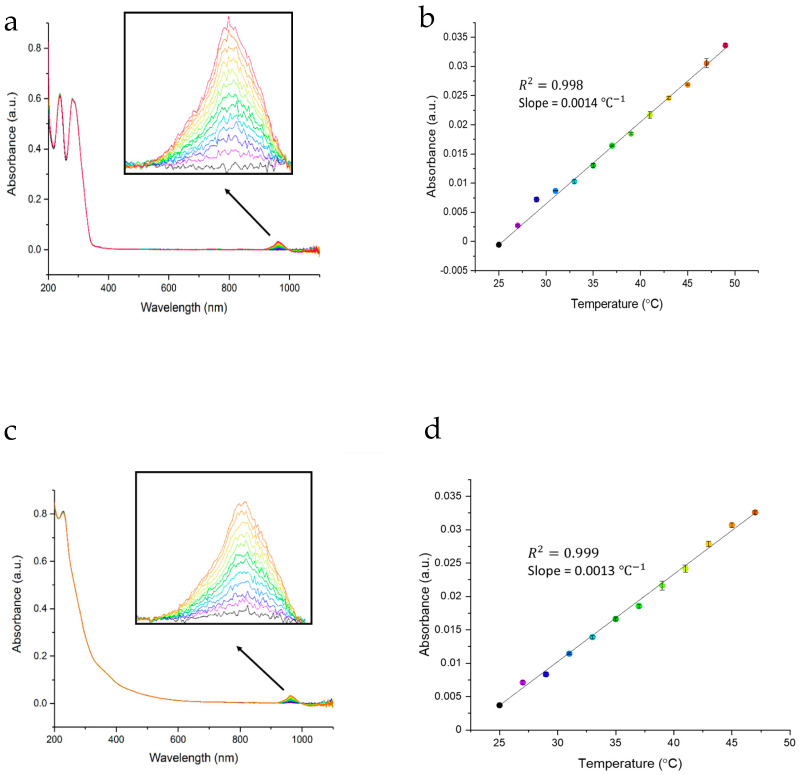
(**a**) N-GQDs absorbance vs. temperature, (**b**) N-GQDs near-infrared spectrum peak absorbance vs. temperature with a linear correlation of R^2^ = 0.998, (**c**) RGQDs absorbance vs. temperature, (**d**) RGQDs near-infrared spectrum peak. absorbance vs. temperature with a linear correlation of R^2^ = 0.999.

**Figure 2 materials-14-00616-f002:**
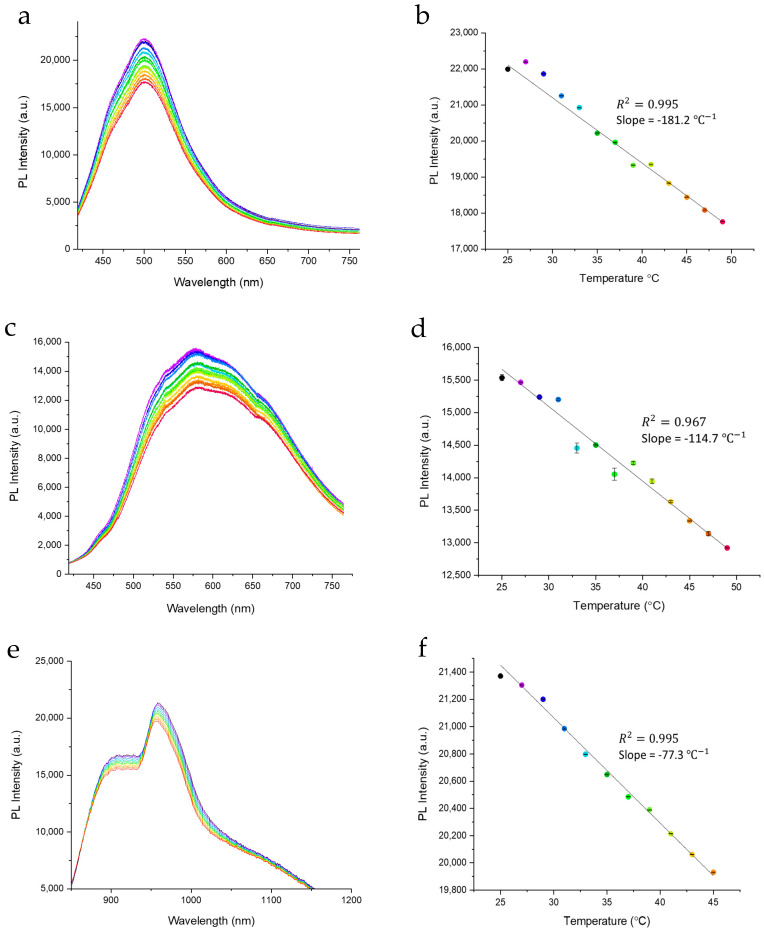
Fluorescence spectral evolution with temperature for (**a**) N-GQD visible emission, (**c**) RGQD visible emission, and (**e**) RGQD near-infrared emission. Linear decrease of peak fluorescence intensity with temperature for (**b**) N-GQD visible emission, (**d**) RGQD visible emission, and (**f**) RGQD near-infrared emission.

**Figure 3 materials-14-00616-f003:**
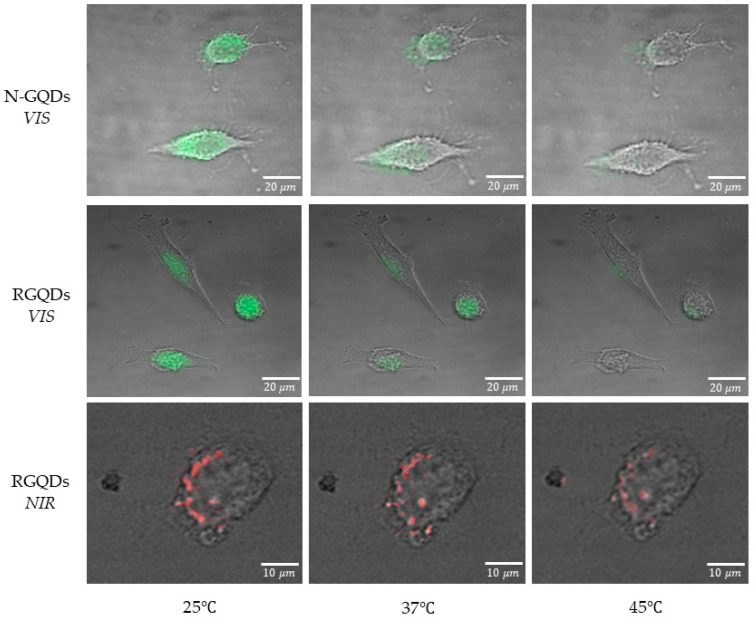
Fluorescence images of N-GQDs and RGQDs internalized into HeLa cells taken at 25, 37 and 45 °C.

**Figure 4 materials-14-00616-f004:**
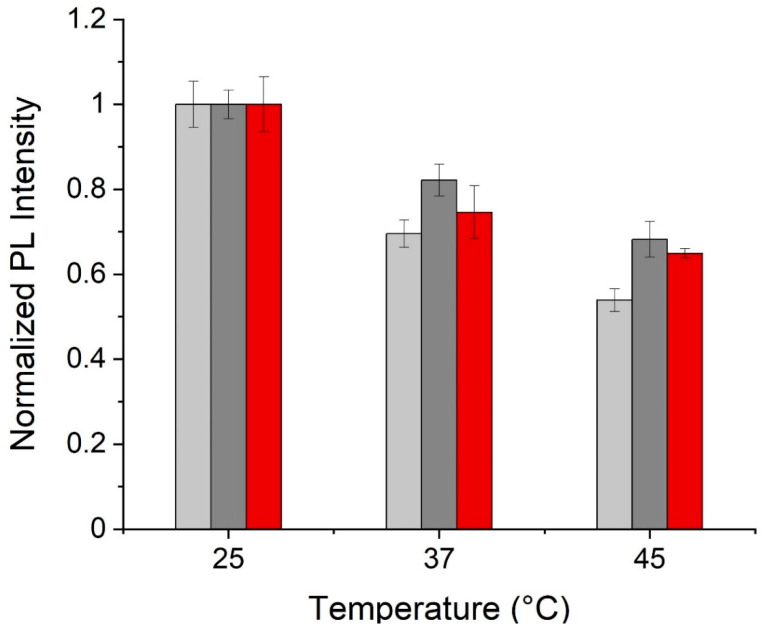
The variation of normalized fluorescence intensity of the GQDs internalized into HeLa cells with temperature assessed for visible fluorescence of N-GQDs (light grey), RGQDs (dark grey) and near-infrared fluorescence of RGQDs (red).

## Data Availability

The data presented in this study are available on request from the corresponding author. The data are not publicly available due to privacy.

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
