# Peer review of "Graphene Quantum Dots as Intracellular Imaging-Based Temperature Sensors"

_materials, 2021, doi:10.3390/ma14030616_

Round 1

Reviewer 1 Report

The authors in this paper use very small and biocompatible nitrogen  doped graphene quantum dots and Reduced graphene  graphene quanum dots for intracellular imaging based temperature sensors. They fluorescence intensity changes with the temperature in the range between 25°C and 45° C. Their results are very interesting to probe cellular environment for a variety of thermodynamics processes in a non invasive way. Their results can open the way for the use of nanothermometry in vivo , in this way can be used together at different techniques such as hyperthermia or photothermal therapy. The paper is well writtenand the text is clear . Their conclusions are consistent with the results obtained and they addressed their question. For this reason the paper for me can be accepted.

Reviewer 2 Report

This paper presents the use of carbon quantum dots for the measurement of temperature in a biologically relevent range using optical techniques.  The chaarterisation of the system is reasonable, there are however a number of papers with very similar appraches.  This  review article might be of use Mohammed, L.J., Omer, K.M. Carbon Dots as New Generation Materials for Nanothermometer: Review. Nanoscale Res Lett 15, 182 (2020).  I think the authors would be able to better justify their novelty by citing more of the relvent works and descibing how their approach differs from those published.  The novelty case could then be strengthened.

In the results section (page 5 , line 191) the mean sizes are presented but the precision is not sensible (sub angstrom) so also present the distrubution (histogram in SI works well).

The text says that the process is reversible, basically saying the temperatures return to the original states; however, the colling curves in the SI are not as linear as those in the heating.  This suggests that there is hysterisis.  I think the authors should be clear about this because the probes are unlikely to be used in the same way as the authors have in this work and thus could create errors.  The gradients of the curves should also be given.

In would be prudent to also present data from the temperature being stepped up and then stepped down in cycles.  This demonstrates reproducability over many heating and cooling cycles.

Fianlly, the problem with any intensity based probe is ensuring you have the same density of particles in the object being measured.  Else, a calibration needs to be performed each time.  Can the authors discuss this limitation and how this can be solved.  For example, some fluorescent probes use lifetime which is not concentration dependent.

Reviewer 3 Report

In this paper, Lee et al. demonstrated Graphene Quantum Dots as Intracellular Imaging-Based Temperature Sensors. It is fascinating work, I hope this article will attract a wide range of audience in quantum dots and biosensing applications. Mainly RGQDs emits lights both visible and NIR wavelength. This paper will be accepted after the following minor issues

  • The quantum yield of the RGQDs shows 8%; how did you measure QY?

Is it in solid form or aqueous solution? Please explain the QY measurement procedure… absolute QY or relative QY measurement?

  • I appreciate the addition of standard deviation bars for all the absorbance or fluorescence quenching data
  • line no. 249 shows high biocompatibility. Did you check any cell viability/toxicity assay?
  • The authors should add the scale bar in the HeLa cell images (Figure.4).
  • Raman spectrum will support your data also with TEM images (Figure S2) of GQDs.
  • I appreciate it if you include the review paper as the 39th reference, elaborately explaining NIR quantum dots for in-vitro and in-vivo applications. DOI: 10.1080/14686996.2019.1590731

Round 2

Reviewer 2 Report

The ammendments have improved the paper for the readership.  I would be happy to see the article published without further corrections.